# The structure of the Vibrio alginolyticus flagellar filament suggests molecular mechanism for the rotation of sheathed flagella

Kailin Qin[1,4] ✉, Rosa Einenkel [2,4], Weilong Zhao [1], Caroline Kühne[2], Joseph Atherton [1], Marc Erhardt [2,3] ✉ & Julien R. C. Bergeron [1] ✉

In several pathogenic bacteria, including *Vibrio* species, the filament of the bacterial flagellum is encased by a membranous sheath, an extension of the bacterial outer membrane. It has been proposed that having sheathed flagella permit bacteria to evade an immune response against flagellar components, suggesting a role in virulence. However, the molecular details of the interaction between sheath and filament, and how it impacts filament rotation, remain largely uncharacterized. Here, we combine single-particle cryo-electron microscopy, cryo-electron tomography, and genetic analyses to resolve the molecular architecture and biogenesis of the sheathed flagellum in *Vibrio alginolyticus*. We show that the flagellar filament forms a canonical 11-stranded supercoil made of the flagellin FlaD2 and enveloped by a bilayered sheath. We report that the filament surface is highly electronegative, suggesting that electrostatic repulsion between filament and sheath may reduce friction and supports high-speed flagellar rotation. We also show that the filament cap protein FliD possesses a unique domain in sheathed flagella, that may coordinate sheath assembly with filament elongation. Collectively, this structural insight into the structure of the *Vibrio alginolyticus* flagellum suggests a molecular mechanism for the rotation of sheathed flagella.

Many bacteria swim through liquid environments using the flagellum. By rotating like a propeller, the flagellum drives bacterial motility and directional movement in many bacterial species[1]. In human gastro-intestinal pathogens, such as *Vibrio cholerae* (*V. cholerae*), *Salmonella enterica* (*S. enterica*), and *Campylobacter jejuni* (*C. jejuni*), the flagellum also plays an important role in invasion and colonization of human tissue[2]. The flagellum consists of a membrane-spanning basal body that houses the export apparatus and motor proteins, and an extra-cellular appendage formed by the hook, the hook-filament junction, the filament, and a cap at the distal end[3,4]. The filament, typically

~10 μm long in most bacteria, is composed of tens of thousands of flagellin subunits – either a single flagellin protein or multiple flagellin variants – which are secreted through the flagellum-specific type-III secretion system[5].

In recent years, cryo-EM structures of flagellum filaments have been reported in many bacterial species. They demonstrate a common architecture, consisting of an 11-stranded filament of the flagellin protein, with slight conformational changes to each subunit allowing the formation of a long-range supercoil. The flagellin consists of two conserved internal domains, D0 and D1, that form the secretion lumen,

[1]Randall Centre for Cell and Molecular Biophysics, King's College London, London, UK. [2]Institute of Biology, Humboldt-Universität zu Berlin, Berlin, Germany. [3]Max Planck Unit for the Science of Pathogens, Berlin, Germany. [4]These authors contributed equally: Kailin Qin, Rosa Einenkel. ✉e-mail: kailin.qin@kcl.ac.uk; marc.erhardt@hu-berlin.de; julien.bergeron@kcl.ac.uk

and several external domains, D2-D5, of which the number and arrangement vary between bacterial species and strains[6,7]. At the tip of the filament, the cap complex, consisting of five copies of the protein FliD, is responsible for flagellin folding and insertion in the growing filament[3,8].

In several bacteria, including the human pathogens *Vibrio* and *Helicobacter*, and the bacterial parasite *Bdellovibrio*, the flagellar filament is surrounded by a membranous sheath, thought to be an extension of the outer membrane[9]. Accordingly, the filament sheath is presumed to be an asymmetric bilayer, containing primarily glycerophospholipids (GPLs) in its inner-leaflet, and lipopolysaccharides (LPS) in its outer-leaflet[10,11], with lipoproteins embedded.

Flagellar sheaths have been proposed to serve multiple, and possibly species-specific, functions[12]. In *Helicobacter pylori*, the sheath has been suggested to protect flagellin from acidic conditions and to contribute to adherence, while in *Vibrio* species, sheaths reduce recognition of flagellin by the host innate immune receptor TLR5 and play a role in outer membrane vesicle (OMV) release[13,14]. Rotation of the sheathed filament has been linked to the production of OMVs, which can mediate host signaling, deliver virulence factors, and provide protection against antimicrobial peptides or bacteriophages. More generally, the sheath has also been hypothesized to shield flagella from flagellotropic phages, which exploit unsheathed filaments for infection[12]. Despite these proposed roles, direct mechanistic insight into the rotation of sheathed flagella has remained limited, with two theoretical models proposed: One where the sheath membrane rotates together with the flagellum filament, and one where only the filament rotates within a flexible sheath.

Low-resolution cryo-ET analyses of sheathed flagella basal body complexes have revealed that these flagella possess additional periplasmic structures, called H-ring and T-ring, thought to promote sheath formation. Deletion of the entire ring structures leads to the formation of periplasmic flagella beneath the peptidoglycan (PG) layer[15]. Some periplasmic filaments penetrate the cell envelope at the region far from the basal body, with and without the sheath. These observations suggest the HT-ring and filament alone play a role in outer membrane remodeling during sheath biogenesis. Moreover, the identification of an extracellular O-ring at the base of the *V. alginolyticus* sheath suggests a further architectural element, although this feature is not conserved across all sheathed species[9]. Despite these observations, the molecular mechanisms underlying sheath architecture remain largely unknown.

Here, we combine single-particle cryo-EM, cryo-ET, and genetic approaches to investigate the structure of sheathed flagella in *Vibrio*. We determine high-resolution structures of both sheathed (3.6 Å) and unsheathed (3.2 Å) filaments and identify FlaD2 as the major polar flagellin of *V. alginolyticus*. Electrostatic analyses reveal strong repulsion between the filament and sheath, consistent with a mechanism that reduces friction during rapid rotation. Furthermore, sequence and functional analyses reveal that the FliD protein possesses an additional domain, which may anchor the flagellum tip to the sheath end. Together, these findings provide structural and mechanistic insights into the adaptations that underlie the rotation mechanism of sheathed flagella.

## Results

### Single-particle analysis of the *V. alginolyticus* sheathed flagellum filament

In sheathed flagella, it has not been shown yet whether the filament forms direct interactions with the membrane. To address this, we analyzed the sheathed filament from *Vibrio alginolyticus* (*V. alginolyticus*), a pathogen affecting a variety of marine animals and humans, causing otitis and wound infection[16,17]. *V. alginolyticus* encodes a single, polar, sheathed flagellum that drives swimming in liquid

environments. In addition, it produces peritrichous, non-sheathed lateral flagella that enable swarming and surface-associated motility under various environmental conditions[18].

To characterize the structure of the *V. alginolyticus* sheathed flagellum, we engineered a mutant of this bacterium lacking the *flhG* gene, previously shown to lead to multi-flagellated bacteria[19]. We subsequently isolated flagellar filaments from this mutant strain using mechanical shearing, achieved by repeatedly forcing a cell suspension through a narrow-gauge needle. Cryo-EM analysis of the resulting sample confirmed that we successfully isolated the flagellar filament with an intact sheath (Fig. 1a). We also obtained sheath-free filaments, as well as filaments exhibiting membrane pearling, suggesting partial detachment of the sheath (Fig. 1b). Using this, we were able to obtain a map of the sheathed filament to 3.6 Å resolution by single-particle analysis (Fig. 1c, d, Supplementary Fig. 1 and Table S1).

This map confirmed that the filament adopts an 11-stranded supercoiled architecture, similar to that of other bacteria. The membrane is well-resolved, with diffused density for the GPL inner leaflet and the LPS outer leaflet clearly identifiable (Fig. 1c, Supplementary Movie 1). Fluorescence microscopy using an outer membrane dye confirmed that in both *V. alginolyticus* and *V. cholerae*, the filament sheath is composed of outer membrane material. This observation is consistent with previous reports[9,20–22]. In contrast, staining with HADA, a fluorescent D-amino acid that labels PG, revealed no signal within the sheath structure. This indicates that the filament sheath is an extension of the outer membrane but lacks an underlying PG layer, distinguishing it from the envelope of the cell body (Supplementary Fig. 2).

### FlaD2 is the main flagellin in *V. alginolyticus*

*V. alginolyticus* encodes six flagellin homologs in its genome (Fig. 2a, Supplementary Fig. 3 and Table S2), FlaB and FlaD1-FlaD5, ~44–99% identical in sequence (FlaD2/FlaD4 are 99% identical). Because of this, it was not immediately clear whether the map of the sheathed filament consisted of a mixture of the above flagellins, or was formed by predominantly one (or a subset of) these flagellins (Fig. 2a). We noted nonetheless that FlaD5 and FlaD1 possess 2–6 residue insertions in their D2 domain, respectively, not present in the other flagellin homologs (Supplementary Fig. 3). In our cryo-EM map of the sheathed filament, there is no density for either of these loops, suggesting that they are not the main flagellins in our structure (Fig. 2b, c). Similarly, in our cryo-EM map, we noted the presence of large side-chain density in positions corresponding to residues Y300, H304, F315, H317 in FlaD2 and FlaD4 (Fig. 2d). In contrast, FlaB, FlaD5, FlaD1, and FlaD3, possess small amino-acids in these positions (Supplementary Fig. 3). Collectively, this suggests that FlaD2/FlaD4 are the main flagellins in the filament map we have obtained.

To verify this, we deleted the genes of all flagellin genes individually and assessed the motility of the corresponding strains. Because *V. alginolyticus* encodes a lateral flagellum system capable of compensating for defects in polar flagellum-dependent motility[18], all experiments were performed in a ΔlafK background (Laf⁻) lacking the lateral flagellar master regulator. Furthermore, a lateral and polar flagellar master regulator mutant was used as a non-motile control strain (ΔlafK ΔflrA). As shown in Fig. 2e, f, deletion of the *flaD2* gene abolished bacterial motility similar to the non-motile control strain, whereas deletion of all the other flagellin homologs led to only a mild (FlaD1) or no significant reduction of motility (FlaD3, FlaD4, FlaD5, FlaB). This confirms that FlaD2 is necessary to form a functional polar flagellum in *V. alginolyticus*. These observations are similar to *V. cholerae*, which also possesses multiple flagellin genes, but a single one (*flaA*) is absolutely necessary for motility in this bacterium (Supplementary Fig. 4a). Whether the various flagellins of *V. alginolyticus* are localized in different zones of the filament, similar to *V. cholerae*, remains to be verified[23].

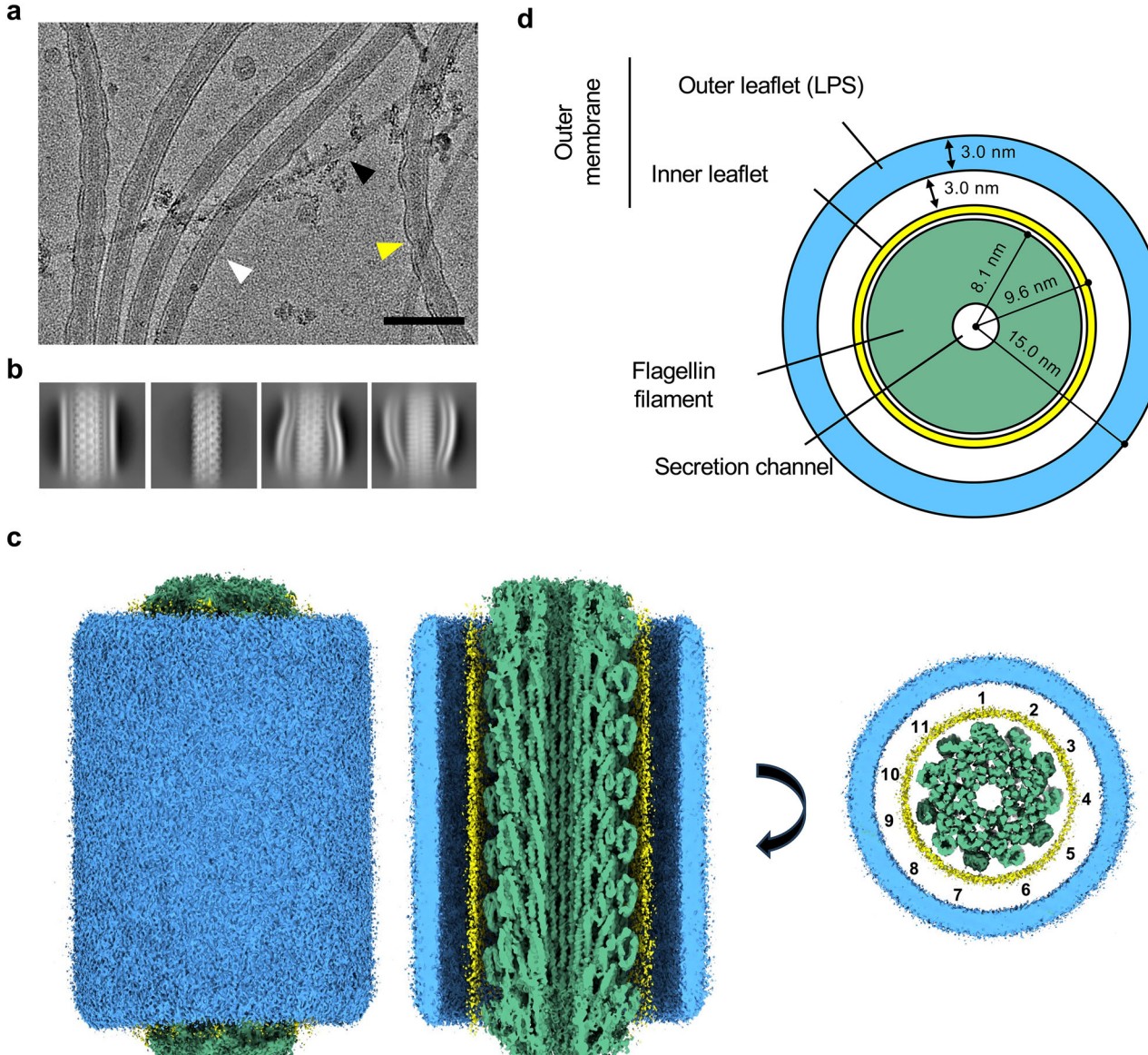

**Fig. 1 | Cryo-EM structure of the sheathed filament of *V. alginolyticus*. a** Representative cryo-electron micrograph (from a 21,747 micrograph dataset) of purified *V. alginolyticus* filaments, with intact sheaths indicated with a white arrow, naked filaments with a black arrow, and pearling sheaths with a yellow arrow. **b** Selected 2D class averages, representing the different types of filaments indicated above. **c** 3D reconstruction of sheathed filaments (left), with cross-sections shown on the right. The sheath outer-leaflet is in blue, its inner leaflet is in yellow, and the filament is in green. 11 proto-filaments are clearly visible. **d** Schematic representation of the sheathed flagellum filament structure, colored as in (**c**).

Collectively, both the motility data and the cryo-EM map reported above suggest that FlaD2 is the main flagellin in the *V. alginolyticus* flagellar filament.

### Structural basis for the filament-membrane interface

Based on the aforementioned results, we built an atomic model of the *V. alginolyticus* filament, using the FlaD2 sequence, in the cryo-EM map of the sheathed filament (Fig. 3a). To further characterize the interaction between the filament and the membrane, we also built an atomic model of the outer membrane, using CHARMM-GUI[24], which could be fitted in the corresponding density (Fig. 3b). In this model, the flagellin outward-most D2 domain faces the membrane inner leaflet, with several residues in proximity (~7–10 Å) to the density attributed to the GPL phosphate groups (Fig. 3c). Intriguingly, these are largely negatively-charged amino-acids, and the intact filament possesses a highly electronegative surface (Fig. 3d). As GPL headgroups are mainly

negatively charged, this raised the question how the contacts between the sheath and the filament are maintained. At this distance, electrostatic forces still exert a large influence[25], which suggests that there is a repulsive force between the sheath and the filament. We postulate that this facilitates the rotation of the filament within the sheath, without any friction that would perturb rotation velocity. In support of this, we note that in most bacteria, the surface of the flagellum filament is largely charge-neutral (Supplementary Fig. 5), supporting the notion that this negative charge of the *V. alginolyticus* filament is a species-specific adaptation permitting to facilitate rotation within a membranous sheath.

### Supercoiling of the *V. alginolyticus* flagellum filament

Previous studies showed that the flagellum filament is supercoiled, with a range of supercoiling architectures observed, all corresponding to subtly different arrangements of the flagellin interactions in the 11

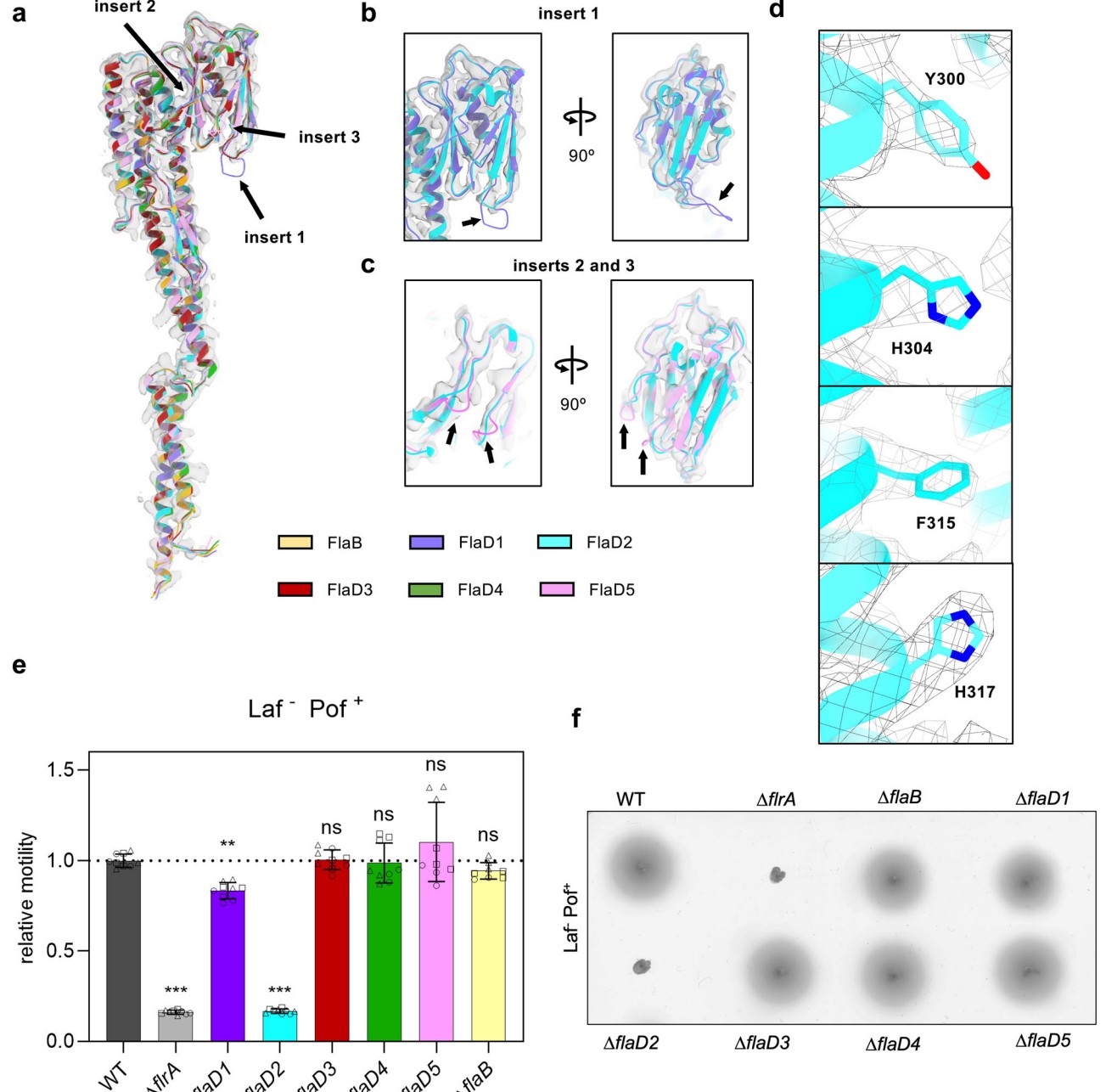

**Fig. 2 | Identification of the flagellin homologs in the *V. alginolyticus* filament structure. a** Structural alignment of the AlphaFold model for all 6 *V. alginolyticus* flagellin proteins. The position of the inserts used for identification is shown with red arrows. **b**, **c** Close-up view of the cryo-EM map of the *V. alginolyticus* filament cryo-EM map, in the positions corresponding to inserts 1, 2 and 3, demonstrating that no density is present for these inserts. **d** Close-up view of the cryo-EM map of the *V. alginolyticus* filament cryo-EM map, in the positions corresponding to residues Y300, H304, F315 and H317, demonstrating the presence of bulky side-chains in these positions. **e** Motility of *V. alginolyticus* strains with individual flagellin genes deleted, in the context of a strain with a defect in the lateral flagella system (Laf⁻).

Only the flaD2 deletion results in abolished motility, suggesting that it is the main flagellin in this bacterium. Bar graphs show mean ± SD from $n = 8$ biological replicates for ΔflaB and $n = 9$ biological replicates for all other mutants with individual data points. Statistical analysis was performed using one-way ANOVA followed by Dunnett's multiple comparisons test (GraphPad Prism). *P*-values are as follows: ΔflrA ***, $p < 0.001$; ΔflaD1 **, $p = 0.002$; ΔflaD2 *** $p < 0.001$; ns, non-significant. Laf-, lateral flagella system defect, Pof+, polar flagella system intact. Source data are provided as a Source Data file. **f** Representative swimming halos for the strains shown in (**e**).

strands of the filament[26]. Supercoiling had previously been reported also in sheathed flagella[23], and indeed, 2D classes of filament particles with a larger box size (Fig. 4a) confirmed that the *V. alginolyticus* sheathed flagellum filament possesses a defined curvature.

Nonetheless, due to the relatively low resolution of the structure of the corresponding filament reported above (Fig. 3a, Supplementary

Fig. 1), caused by the limited number of particles, we were not able to resolve the differences between flagellin subunits leading to this curvature. To address this, we determined the structure of the unsheathed filament, from the corresponding particles present in the aforementioned dataset, to 3.16 Å resolution (Supplementary Fig. 1, Supplementary Fig. 6a). In this structure, we were able to build 55 copies of the FlaD2 flagellin.

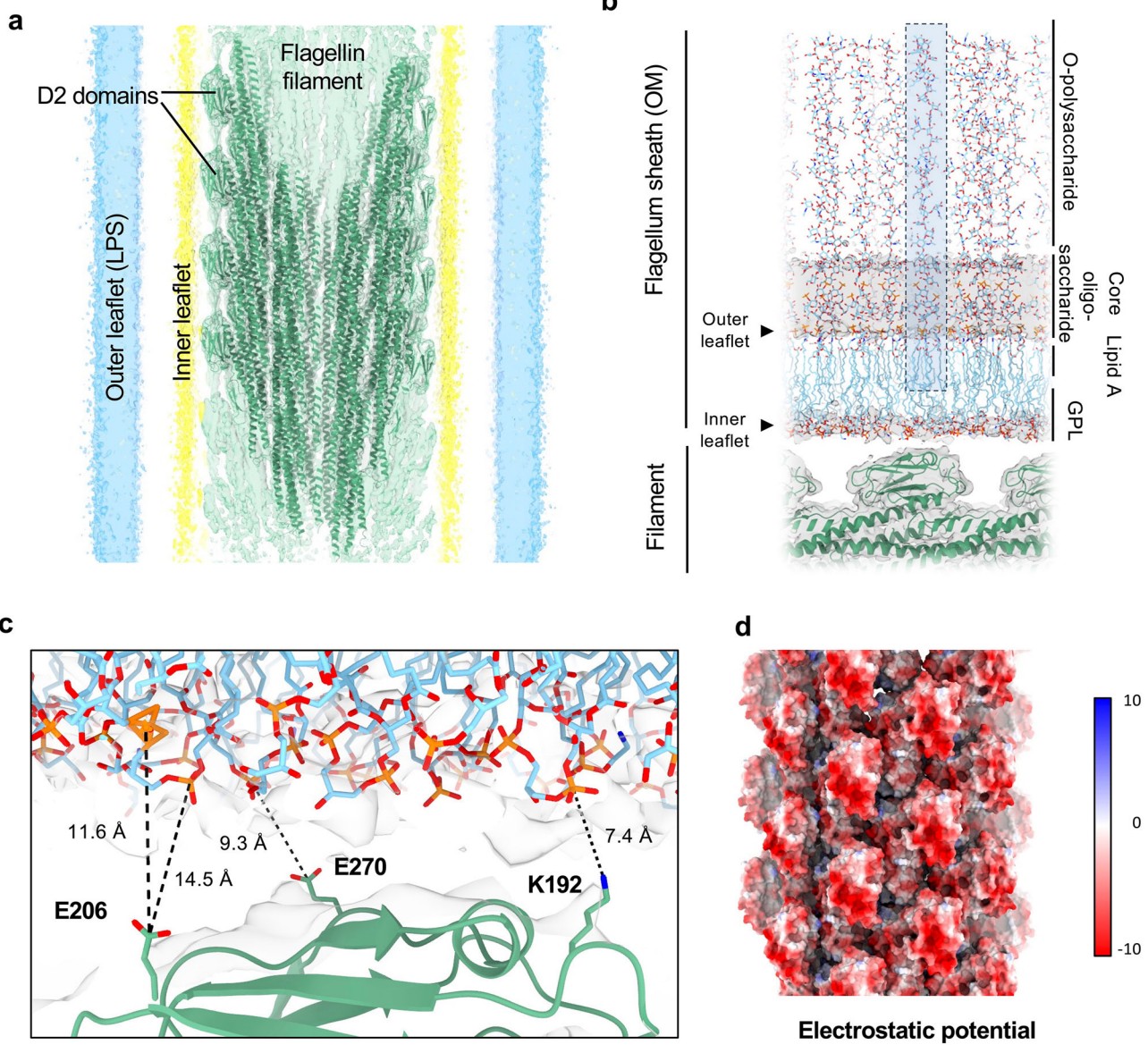

**Fig. 3 | Structural basis for the filament-membrane interface. a** Atomic model of the *V. alginolyticus* flagellum filament, fitted into the cryo-EM derived map. Elements are colored as in Fig. 2. **b** Close-up view of the corresponding map, with a model of the bacterial outer membrane built in. **c** Electrostatic surface representation (in kT/e) of the *V. alginolyticus* flagellum filament. The surface is highly electronegative. **d** Close-up view of the interface between FlaD2 and the sheath.

Both structures (sheathed and unsheathed) are highly similar, and notably we observed no significant structural changes in the D2 domain (Supplementary Fig. 6b). This also confirmed that this structure corresponds primarily to FlaD2 filaments lacking the sheath, not to filaments formed by other polar flagellins, nor that of the lateral flagellum, for which the flagellin LafA shares only ~52% identity with FlaD2. Thus, the isolated unsheathed filaments likely originated from the polar flagellum, from unsheathed filaments in *V. alginolyticus*[9] or from filaments that lost their sheath during isolation.

In the resulting structure, obtained without helical symmetry imposed, we could clearly distinguish convex and concave sides on the filament (Fig. 4b), in agreement with the 2D classes described above. Comparison of individual flagellin protofilaments revealed that each possesses a distinct curvature, with the protofilament from the concave side significantly more curved (Fig. 4c). A close-up view of the longitudinal interfaces of protofilaments from the concave and convex sides revealed subtle differences in the interaction (Fig. 4d), which

explain the difference in curvature when propagated across many subunits.

These observations confirmed that the supercoiled nature of the flagellum filament is also conserved in sheathed flagella.

### The cap protein FliD possesses a unique domain, likely anchored to the sheath

The flagellum cap protein FliD is required for filament formation in most bacteria, as it promotes flagellin folding during filament assembly. Our previous structural studies had revealed that FliD forms a pentameric structure at the tip of the filament, with domains D0-D1 buried in the filament structure, and D2-D3 forming the pentameric contacts at the filament tip[3,8]. Intriguingly, analysis of FliD sequences revealed that in both *Vibrio* spp and *Helicobacter* spp, FliD possesses a distinct architecture, with predicted unique domains (termed D4 in *Vibrio*, and D4-D5 in *Helicobacter*) not found in other FliD orthologues (Supplementary Fig. 7a). Modeling of the *V. alginolyticus* D4 domain

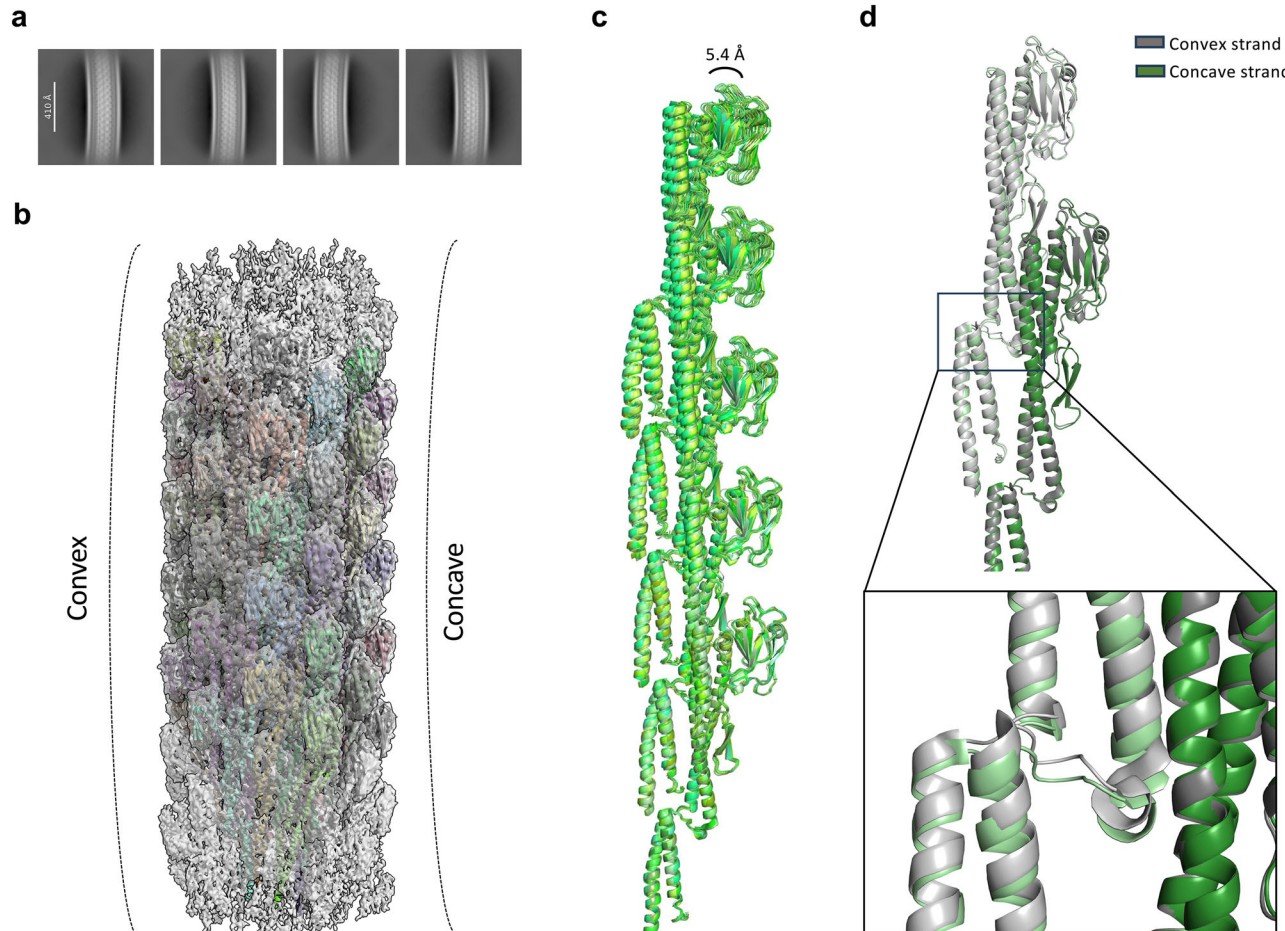

**Fig. 4 | Supercoiling of the *V. alginolyticus* flagellum filament. a** Selected 2D class averages, representing sheathed filaments with a defined curvature. **b** Atomic model of the *V. alginolyticus* unsheathed flagellum filament, fitted into the cryo-EM derived map showing convex and concave sides. **c** Comparison of individual flagellin protofilaments reveals that the protofilament from the concave side is significantly more curved. **d** Close-up view of the longitudinal interfaces of protofilaments from the concave and convex sides.

with AlphaFold3[27] revealed that it consists of two helical domains, located on top of the cap complex (Supplementary Fig. 7b). This allowed us to build a hybrid model of the *V. alginolyticus* filament-cap complex (Fig. 5a, Supplementary Fig. 7c), by combining this model to our previously-published structure of the *S. enterica* cap-filament complex[8], and the structure of the sheathed filament reported above. Notably, the diameter of the resulting cap complex is ~16.8 nm, larger than that of the filament (Supplementary Fig. 7c), which is similar to the width of the sheath inner leaflet. This suggests that the edge of the FliD D4 domain could be directly interacting with the sheath itself (Supplementary Fig. 7d).

To verify this, we constructed mutant strains lacking the D4 domain in FliD (*fliD*$_{\Delta D4}$). As shown in Fig. 5b, negative-stain TEM analysis of this mutation in *V. cholerae* revealed the presence of empty sheath extensions at the tip of approximately one in four filaments (3 observed in 13 cells), not observed in WT bacteria (0 observed in 15 cells). This observation suggests a role of the FliD D4 domain in synchronizing filament assembly with sheath elongation. We note, however, that *fliD*$_{\Delta D4}$ strains retained swimming motility in soft agar, in both *V. alginolyticus* and *V. cholerae*, in contrast to the respective Δ*fliD* deletion mutants (Supplementary Fig. 8), indicating that the D4 domain of FliD is not essential for filament assembly.

To further investigate the architecture of the *V. alginolyticus* cap complex in-situ, we employed cryo-electron tomography (cryo-ET) to visualize filament tips in the *V. alginolyticus* Δ*flhG* mutant described

above. As shown in Fig. 5c, Supplementary Movie 2, the distal tips of the flagellar filaments were clearly resolved in the tomograms, and the surrounding sheath appeared continuous and sealed, with no detectable gap or bulge at the filament tip. Structural docking of the filament–cap complex model into these tomograms supports the presence of an intact cap structure beneath the closed sheath (Fig. 4d, Supplementary Movie 2). Collectively, these results indicate that while the FliD D4 domain is not essential for sheath formation or maintenance, it may contribute to the coordination between sheath elongation and filament growth. The molecular basis of this contribution remains to be clarified.

Finally, we sought to characterize the architecture of the cap complex in the early stages of flagellum assembly. To this end, we engineered a *V. alginolyticus* Δ*flhG* Δ*fliS* mutant, in which secretion of flagellin and therefore motility is strongly reduced (Supplementary Fig. 8a)[28], and used cryo-ET to gain structural insights into their flagellum architecture. As expected, flagella were significantly shorter in this mutant, yet the sheath was present and still closed at the tip (Supplementary Fig. 9a). Intriguingly, we observed a range of flagellar architectures (Supplementary Fig. 9b), including ones where the cap complex is near the tip of the sheath (5 out of 9 flagella observed), and ones where the filament is significantly shorter, with the cap complex visible below the tip of the sheath (3 out of 9 flagella). We also observed one instance of an empty sheath without a filament. These results suggest that synchronization between filament assembly and sheath

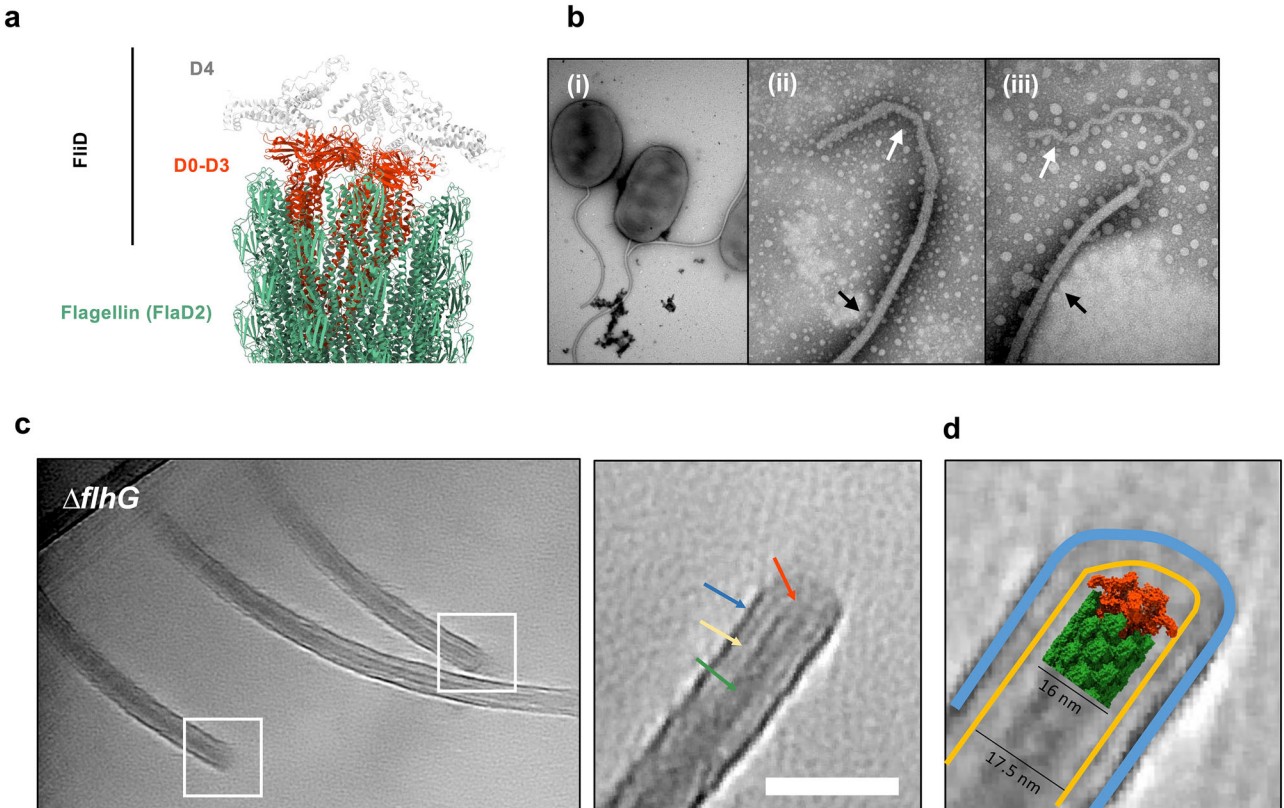

**Fig. 5 | A domain in the cap protein FliD anchors the flagellum filament to the sheath. a** Atomic model of the *V. alginolyticus* filament-cap complex. **b** Representative negative-stain TEM micrographs (from a 10-micrograph dataset) of *V. cholerae* cells (i) and filament tips (ii), (iii) for the *fliD*$_{\Delta D4}$ mutant. The sheath expands past the filament tip, demonstrating a lack of coordination between filament and sheath assembly in this mutant. Arrows indicate the empty sheath (white) and the sheathed filament (black). **c** Representative cryo-electron tomogram (from 4 reconstituted tomograms containing flagella) of filament tips in the hyper-flagellated strain (*ΔflhG*) of *V. alginolyticus*. Tips are indicated with a white box, with a selected close-up view shown on the right. Arrows indicate the membrane outer later (blue), inner layer (yellow), filament (green) and cap (orange). Scale bar: 50 nm. **d** Schematic representation of sheath tip architecture. The atomic model of the filament-cap complex was positioned at the correct location in the sub-tomogram, with the membrane leaflets segmented in blue (outer leaflet) and yellow (inner leaflet). The FliD D4 domain has a similar diameter to the inner leaflet.

elongation occurs later during flagellum biogenesis, although we cannot rule out that this is an artifact caused by the *ΔflhG ΔfliS* mutation, which significantly disrupts flagellum assembly and regulation.

## Discussion

In this study, we present the structure of the sheathed flagellum filament in *V. alginolyticus*, providing insights into the interface between the filament and the membranous sheath. Based on sequences of flagellin homologs and EM densities of the filament, we demonstrate that FlaD2 is the main flagellin in this bacterium. Furthermore, electrostatic analyses indicate a uniformly negatively charged filament surface, presumably strongly repulsive between the filament surface and the sheath, which may help to reduce friction and thereby facilitate rapid rotation of the filament within the sheath. Finally, we report that the cap complex possesses a unique architecture in sheathed flagella, which is not required for filament assembly but may contribute to synchronizing it to sheath elongation.

Based on these results, we propose that for sheathed flagella, there is no direct molecular interaction between the sheath and the filament during flagella rotation. Nonetheless, our data suggest that the cap complex is forming direct contact with the sheath inner layer during flagella assembly to synchronize filament growth with sheath elongation (Fig. 6a). A recent study examining the architecture of the sheathed flagellar motor of *V. cholerae* likewise proposes that sheath biogenesis is coordinated with ongoing flagellar assembly, specifically the coordination between HL-ring formation and hook elongation[23]. This complementary perspective, though focused on distinct regions of the flagellum, supports the broader view that sheath growth and flagellar assembly are mechanistically linked.

In this model, we propose that charge repulsion "greases" the interface between sheath and filament, ensuring that the filament rotates within the sheath without friction (Fig. 6b). Together with the previously reported continuity with the outer membrane, and the lack of direct contacts between the filament and sheath, this observation provides molecular support for earlier models proposing that the filament rotates freely within a flexible sheath, rather than as a rigid filament-sheath complex[23,29]. Such a mechanism would allow the sheath to deform in response to filament motion while maintaining structural integrity.

It remains to be determined how the sheath maintains its tubular structure, in the context of surrounding a filament with repulsive electrostatic interactions. It has been shown that the outer leaflet of the bacterial outer membrane is relatively rigid[30], and it is plausible that the LPS structure alone is sufficient to maintain a tubular sheath. Nonetheless, further molecular and computational studies will be required to confirm this hypothesis.

We also note that rotation of sheathed flagella in *Vibrio* has been shown to promote the release of outer membrane vesicles (OMVs), which contribute to immune modulation, host colonization, and delivery of virulence factors[13,31]. Electrostatic repulsion could facilitate

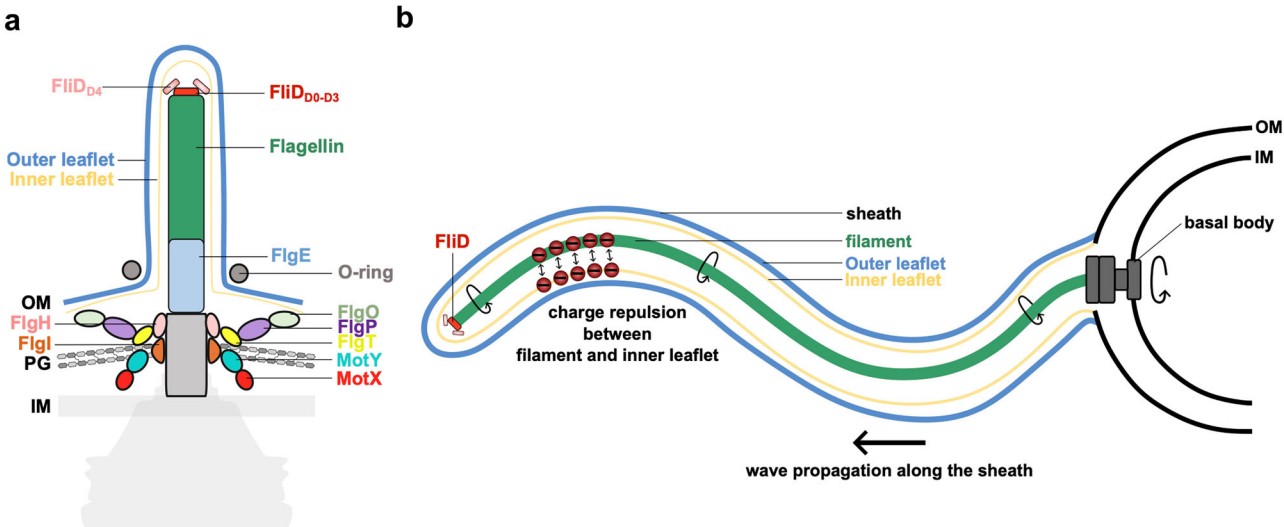

**Fig. 6 | Schematic representation of the proposed sheathed filament structure, and molecular mechanism for its rotation. a** Schematic representation of the sheathed flagellum of *V. alginolyticus* showing the proposed interactions of the FliD D4 domain with the inner leaflet of the sheath. **b** Proposed molecular mechanism of filament rotation within the membranous sheath. Charge repulsion between the filament and the sheath facilitates the rapid rotation of the filament within the sheath.

local membrane curvature and blebbing, consistent with the bubbling we observed in our cryo-EM datasets (Fig. 1a, b). Although speculative, these observations suggest a potential link between sheath architecture, filament-sheath interactions, and OMV biogenesis that warrants future investigation.

Taken together, our structural and genetic analyses provide a framework for understanding how sheath architecture is supported in *Vibrio* and highlight the unique adaptations of sheathed flagellar filaments compared to unsheathed systems. These findings establish a basis for future work to dissect the molecular mechanisms of sheath formation and its impact on bacterial motility and virulence.

## Methods

### Bacterial strains and growth conditions
All *Vibrio alginolyticus*, *Vibrio cholerae*, and *Escherichia coli* (*E. coli*) strains used in this study are listed in Table S3. The strains were derived from *V. alginolyticus* NCTC 10675 or *V. cholerae* V52 Δ*vasK*[32]. The wild-type *V. alginolyticus* NCTC 10675 was obtained from the National Collection (UK Health Security Agency, UK) and originally isolated by Miyamoto et al.[33].

Bacteria were cultured in lysogeny broth (LB) at 37 °C with constant shaking at 180 rpm. For *V. alginolyticus*, LB was supplemented with 2% NaCl (LBS). Antibiotics were added as required at the following final concentrations: chloramphenicol (6.25 μg/ml for *V. alginolyticus*, 12.5 μg/ml for *V. cholerae* and *E. coli*), ampicillin (50 μg/ml), and streptomycin (100 μg/ml). Bacterial growth was monitored by measuring optical density at 600 nm (OD$_{600}$) using a spectrophotometer (Amersham Bioscience).

### Genetic manipulations
Chromosomal mutations were introduced into the genomes of *V. alginolyticus* and *V. cholerae* using the suicide plasmid pRE112, which carries the *sacB* gene for sucrose-based counterselection[34]. Approximately 600 bp of DNA up- and downstream of the target gene were amplified and assembled into pRE112 using NEBuilder HiFi DNA Assembly Master Mix (New England Biolabs, catalog number: E2621L), then transformed into *E. coli* DH5α λ-pir⁺ or *E. coli* CC118 λ-pir⁺, and after confirmation of correct assembly into *E. coli* SM10 λ-pir⁺. Conjugation was performed between donor *E. coli* SM10 λ-pir⁺ and

recipient *Vibrio* strains. Recombinants resulting from single crossover events were selected on LB/LBS plates containing chloramphenicol, supplemented with ampicillin (*V. alginolyticus*) or streptomycin (*V. cholerae*). To select for the second crossover and excision of the plasmid backbone, cultures were grown for 4 h in LB/LBS, and plated on no-salt LB agar supplemented with 10% sucrose and streptomycin (*V. cholerae*) or on tryptic soy agar (TSA) supplemented with 10% sucrose and ampicillin (*V. alginolyticus*). Colonies were screened for loss of chloramphenicol resistance to confirm plasmid excision, and the desired chromosomal modifications were verified by PCR and Sanger sequencing. Lists of plasmids and oligonucleotides used in strain construction are provided in Tables S4, S5.

### Isolation of flagellar filaments
A single colony of wild-type *V. alginolyticus* was inoculated into 5 ml LB medium and cultured overnight at 30 °C. The next day, 250 mL LB medium was inoculated 1:100 with the overnight culture and grown to OD 0.5–0.8. Cells were harvested at 4000 × *g* at 4 °C for 10 min. The cell pellet was resuspended in 40 mL ice-cold sucrose buffer (20 mM Tris-HCl, 20% w/v sucrose, pH 8.0) on ice. Flagellar filaments were sheared off by passing the cell-suspension through a 23 ga, 0.7 × 50 mm needle using a 20 mL syringe 10 times. To separate the cells from the sheared filaments, the cell suspension was centrifuged at 20,000 × *g* at 4 °C for 30 min, and the supernatant containing the sheared filaments was collected and centrifuged at 100,000 × *g* at 4 °C for 1 h. The filaments in the pellet were resuspended in 200 μL ice-cold sucrose solution (20 mM Tris-HCl, 20% w/v sucrose, pH 8.0).

### Single-particle cryo-EM sample preparation, data acquisition, and processing
Filament resuspension (3 μL) was applied to glow-discharged holey carbon grids (Quantifoil R2/2, 300 mesh). Samples were incubated for 30 s at 4 °C and 88% humidity before being blotted by Leica EM GP1 and then rapidly plunged into liquid ethane. Grids that were blotted for 5–8 s were screened on a 200 kV Glacios microscope (Thermo Fisher). The grids with good ice thickness were deposited to 300 kV Krios G3i microscope with a Gatan K3 direct electron detector (Thermo Fisher). The dataset was collected using a physical pixel size of 1.078 Å at a magnification of 81,000 ×. Finally, 21,747 movies were collected with a

total dose rate of 40 e-/Å², with 40 frame fractionations. All movies were collected over 40 frames with a defocus range of −0.9 μm to −2.7 μm.

For all movies, motion-correction and CTF estimation were processed in CryoSPARC v4.7[35] using patch motion correction and patch CTF estimation, respectively. A total of 21,634 movies were used. Filament tracer was performed for particle picking, with a filament diameter of 200 Å and a separation distance of 0.265-fold diameters. 3,059,499 particles were extracted with a box size of 500 × 500 pixels and subjected to 2D classification. Two rounds of 2D classification were performed, to remove junk particles and to separate sheathed and unsheathed filament, respectively. For the sheathed filament, a subset of 76,818 particles were subjected to a round of helical refinement, followed by a round of local CTF refinement. Two rounds of 3D classification were performed to remove particles that present abnormal sheaths. Finally, a local refinement with a tight mask was performed with 72,779 particles and a map with a resolution of 3.6 Å was obtained. For the unsheathed filament, a subset of 443,577 particles was subjected to a round of helical refinement, followed by local CTF refinement. A round of 3D classification was performed to remove junk particles. Finally, a local refinement with a tight mask was performed with 439,378 particles and a map with a resolution of 3.16 Å was obtained.

## Model building and refinement
Atomic models for all *V. alginolyticus* NCTC 10675 flagellin homologs, namely FlaD1, FlaD2, FlaD3, FlaD4, FlaD5, were obtained from the AlphaFold database[27], and fitted in the cryo-EM maps. Based on the density, we identified FlaD2 as the main flagellin in the structure (see results). Therefore, we used FlaD2 to build the atomic models for both sheathed and unsheathed filaments. 33 flagellin monomers (sheathed filament) or 55 flagellin monomers (unsheathed) were manually fitted into the reconstructed maps, and the complex was flexibly refined by ISOLDE[36] with secondary structure restraints gained from the Alpha-Fold model in UCSF ChimeraX[37]. Real-space refinement in PHENIX[38] with secondary structure, rotamer and Ramachandran restraints but without NCS restraints was next employed. Coot[39] was then used to correct rotamer outliers, side-chain clashes, and unattributed density. The final model was validated using the validation program in PHENIX[38].

The atomic model of the bacterial outer membrane was generated with the membrane builder in CHARMM-GUI[24], using a 1:1:1 ratio of phosphatidylethanolamine: phosphatidylglycerol: cardiolipin for the inner leaflet, and LPS for the outer leaflet.

Atomic models of the *V. alginolyticus*, *V. cholerae*, and *H. pylori* FliD orthologues were obtained from the AlphaFold database. A model of the *V. alginolyticus* cap-filament complex was generated by aligning five copies of the *V. alginolyticus* FliD model, and the structure of the *V. alginolyticus* filament (this study), onto the structure of the *S. enterica* filament-cap complex; this model was fitted in the tomogram of filament tips using ChimeraX[37].

## Cryo-ET sample preparation, data collection and processing
Overnight cultures of the respective strains were diluted 1:100 in 10 mL LBS and incubated for 1.5 h. Cells were centrifuged at 2500 × *g* for 10 min and resuspended in ice-cold sucrose buffer (20 mM Tris-HCl, 20% w/v sucrose, pH 8.0) in a gentle manner. The pipette tip was cut off to reduce the flagella to be sheared off. 4 μL cell suspension was applied to glow-discharged lacey carbon grids (Agar Scientific, 200 mesh). Samples were incubated for 30 s at 5 °C and 95% humidity before being blotted by Vitrobot Mark IV and then rapidly plunged into liquid ethane. Grids that were blotted for 8, 12, 16 and 20 s were screened on a 200 kV Glacios microscope (Thermo Fisher). Cryo-electron tomography (cryo-ET) data were collected using a Titan Krios TEM (Thermo Fisher) operated at 300 kV and equipped with a K3

direct camera (. Tomograms were collected using the TOMO5 software (Thermo Fisher) with a pixel size of 2.1 Å at 46,000 x. 21 tilt series ranging from −50 to +50 degrees with an increment of 5 degree and a fixed defocus of −3.5 μm were collect for each position in dose-symmetric tilt scheme (36 tilt series for the ΔflhG mutant, and 84 tilt series for the ΔflhG ΔfliS mutant). A dose of 1.5 ‐ 2.0 e‐/Å‐² per tilt was set, resulting in a total dose of 30 ‐ 40 e‐/Å‐². Relion 5[40] was used for motion correction, CTF correction, tilt series alignment (AreTomo3), tomogram reconstruction, and tomogram denoising (cryoCARE).

## Motility assay
Swimming motility was studied using tryptone broth-based soft agar swim plates containing 0.3% Bacto agar, supplemented with 2% NaCl for *V. alginolyticus*. Motility plates were inoculated with 2 μL of overnight culture and incubated at 37 °C for 6–8 h. Images were acquired by scanning the plates, and the diameters of the swimming halos were measured using Fiji[41]. The swimming diameters of the mutant strains were normalized to those of the respective WT.

## Outer membrane and peptidoglycan staining
Overnight cultures were diluted 1:100 into 5 mL M9 minimal medium supplemented with 0.4% glucose, 2 mM MgSO₄, 0.1 mM CaCl₂ and 2% NaCl. For each strain, 500 μL aliquots were transferred into a new tube, while the remaining 4.5 mL culture was used to monitor growth at 37 °C, 180 rpm. To perform dual labeling, HADA (Tocris BioScience, catalog number: 6647/5) was added to the culture to a final concentration of 500 μM. Cells were incubated for 4–5 h at 37 °C in the dark with shaking at 650 rpm. Thirty minutes before harvesting, FM1-43 (Invitrogen, catalog number: T3163) was added to a final concentration of 15 μg/mL. After staining, 1 mL of fresh M9 medium was added to dilute the stains. Cells were pelleted by centrifugation at 2500 × *g* for 5 min, and the supernatant was discarded. Pellets were washed twice: first resuspended in 1 mL of fresh medium, pelleted again at 2500 × *g* for 2 min, and then resuspended in 500 μL of the same medium. The samples were applied to home-made flow cells prepared with poly-L-lysine (PLL)-coated coverslips that were prepared as described previously[42]. Briefly, coverslips were incubated with 0.1% PLL for 10 min, then air-dried, and subsequently fixed to an objective slide via two layers of pre-heated parafilm to create a chamber. The side of the coverslip incubated with PLL faced the objective slide. Cells were allowed to adhere for 10 min in the dark in an inverted position. Non-adherent cells were gently washed off by rinsing twice with 40 μl of M9 medium. Mounting medium (Fluoroshield, Sigma Aldrich, catalog number: F6182) was added before imaging. Fluorescence microscopy was performed using a Ti-2 Nikon inverted microscope equipped with a CFI Plan Apochromat DM 60× Lambda Ph3/1.40 (Nikon) oil objective, an Orca Fusion BT camera (Hamamatsu), and a SPECTRA III LED light source (Lumencor). Z-stack images were acquired every 0.4 μm across a 1.6 μm range (5 slices). FM1-43 was excited with a 488 nm laser at 20% power and 100 ms exposure. HADA was excited with a 365 nm laser at 20% power and 100 ms exposure. Emission was collected using the GFP emission filter (499–530 nm; FF01-515/30, Em2) and DAPI emission filter (414–450 nm; FF01-432/36, Em1) from the LED-DA/FI/TR/Cy5/Cy7-A Full Multiband Penta filter (Semrock, IDEX). Images were analysed using Fiji[41].

## Statistical analyses
Statistical analyses were performed using GraphPad Prism 10 (Graph-Pad Software, Inc., San Diego, CA), and values of $P < 0.05$ were considered statistically significant.

## Molecular modeling
A homology model of the *V. alginolyticus* FliD pentamer was built with SwissModel, with the *S. enterica* structure (PDB ID: 9GNZ) as a template. The structure of the *V. alginolyticus* FliD D3-D4 region was

modeled using the AlphaFold3 server[27], and this model was added to each FliD chain, aligned on the D3 domain.

For the full filament-cap model, the FliD pentamer model was aligned to the *S. enterica* FliD-filament structure (PDB ID: 9GNZ), and the *V. alginolyticus* filament structure (See above) was aligned to the *S. enterica* filament in the same structure.

## Reporting summary

Further information on research design is available in the Nature Portfolio Reporting Summary linked to this article.

## Data availability

The coordinates and/or EM maps that are displayed in this paper have been deposited to the PDB and/or EMDB databases with the following accession codes: sheathed filament, PDB: 9RCD, EMDB: EMD-53917; unsheathed filament, PDB: 9RCB, EMDB: EMD-53912; tomogram of multiple flagella tips in *ΔflhG*: EMD-53992; tomogram of multiple short flagella in *ΔflhG ΔfliS*: EMD-53993. Source data are provided with this paper.

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

## Acknowledgments

K.Q. and W. Z. are supported by a PhD studentship from the China Scholarship Council, and K. Q. acknowledges the Randall Center for supplementary funding towards PhD bench fees. M.E. acknowledges funding from the European Research Council (ERC) under the European Union's Horizon 2020 research and innovation program (grant agreement n 864971) and from the Max Planck Society as a Max Planck Fellow. J.R.C.B. acknowledges funding from the BBSRC (BB/R009759/2) and HFSP program (RGY0080/2021). We thank members of the Erhardt and Bergeron labs for helpful discussions. We thank Christian Goosmann (Max Planck Institute for Infection Biology) for TEM grid preparations and observation of *Vibrio* cells. Cryo-EM grids were screened at the Imperial College London cryo-EM facility (funded by BBSRC grant BB/V019732/1), and data were collected at the LonCEM facility (funded by Wellcome Trust grant 206175/Z/17/Z); we acknowledge Paul Simpson and Nora Cronin, respectively, for support.

## Author contributions

J.R.C.B., M.E., and K.Q. conceptualized the research project, and J.R.C.B. and M.E. ensured funding. R.E. generated chromosomal *V. cholerae and V. alginolyticus* mutants, performed and analysed motility assays and fluorescent microscopy experiments on mutants with the help of C.K. K.Q. prepared negative-staining grids and collected TEM images for *V. cholerae*. K.Q. isolated the flagella of *V. alginolyticus*, prepared the EM grids and collected and processed the EM data. K.Q. reconstructed and refined the EM map of the sheathed and unsheathed filament, and built and refined their atomic models. For tomography on *Vibrio* flagella, K.Q. collected the tomography data, and W.Z. processed the data, with support from J.A. J.R.C.B., K.Q., and R.E. wrote the first draft of the manuscript, with comments from all authors. K.Q. and R.E. prepared figures and K.Q. prepared movies. M.E. and J.R.C.B. reviewed and edited the manuscript. All authors reviewed the results and approved the final version of the manuscript.

## Competing interests

The authors declare no competing interests.
