## [Transparent Peer Review file · Nature Communications]

The structure of the *Vibrio alginolyticus* flagellar filament suggests molecular mechanism for the rotation of sheathed flagella

Corresponding Author: Dr Julien Bergeron

Version 0:

Reviewer comments:

Reviewer #1

(Remarks to the Author)

I have reviewed this manuscript before for another journal. This version is improved and the authors have satisfactorily answered my previous comments.

Only one point to highlight, I did not follow the authors' answer to point 1 (Reviewer 1). First, line 276 is on page 9 in the version I have and it has nothing to do with the point being raised here (it corresponds to the third line of the discussion in the version I have). Second, I would strongly encourage the authors to mention the statistics of each observation they describe in the paper, or just make a supplementary table summarizing the number of examples related to each observation they describe and the number of cryo-tomograms. For example, "we looked at XX number of WT *V. cholerae* with TEM and we did not observe empty sheath tips in the filaments", "we looked at XX number of FliDd4 mutant and XX% of them had empty sheath tips", "we collected XX cryo-tomograms of *V. alginolyticus* flhGfliS mutant and saw this number of empty OM extensions", same applies for WT, etc.

Reviewer #2

(Remarks to the Author)

The authors have substantially rewritten their manuscript and resolved my earlier concerns about overstatements in the interpretation of their results. What remains is solid work, in part confirming earlier findings (e.g. membrane staining of sheath and lack of PG under the sheath was previously established), and in part advancing the field (structure of flagellin, and relevance of FliD D4 for interaction between sheath and filament at tip).

Remaining comments:

1. I agree with reviewer 1 that the authors should give some statistical indication of how common empty sheaths are in the Delta FliD D4 mutant compared to the WT. It is helpful that the manuscript now states that these were not observed at all in the WT, but it would also be helpful if there was information on how common they are in the mutant (how many of how many inspected sheaths seemed empty?).
2. Is it possible to infer anything about interactions between inner membrane and FliD D4 based on the charge of the latter? Minimally, one would expect that D4 should be less negatively charged than the surface of the flagellin to permit a favorable interaction.

Minor comments:

1. References appear to be missing for a number of statements if not whole paragraphs, e.g.

- 1.1. I 52-60
- 1.2. I. 189-191
- 1.3. I. 233
- 2. I. 61-62: Bdellovibrio is not a human pathogen as stated.

Reviewer #3

(Remarks to the Author)

The revised manuscript by Qin et al., now titled "The structure of the *Vibrio alginolyticus* flagellar filament suggests molecular mechanism for the rotation of sheathed flagella," has been reformatted as a regular article for Nature communications. The authors' data demonstrate: (1) a structure of the *V. alginolyticus* flagellar filament obtained by cryoEM and cryoET; (2) that FlaD2 is the dominant flagellin in the filament; (3) the physical arrangement of flagellin D2 into protofilaments; (4) how helical interactions between protofilaments generate 11-fold symmetry; (5) that the filament is curved and has a negatively charged exterior; and (6) a tentative assignment for the cap at the flagellar tip. I was the original Reviewer 3 and had initially assumed that some of the claims that did not seem fully supported by the data might reflect the condensed format, and the original review focused on how the selected format limited clarity. I also made comments on additional metrics that would be useful in the structural table, which have been addressed.

In the revised manuscript, which now uses a longer format, the lack of clarity remains (unfortunately) as do the speculations. In terms of clarity, the manuscript is not sufficiently cohesive, and the wording often lacks scientific precision (the other reviewers gave specific examples of lack of precision, but I felt that this was extensive and influenced by the format. It remains extensive and beyond what is reasonable for a reviewer to list one by one). The narrative at times shifts between first- and third-person voice and between active and passive voice. It sometimes conflates introduction/results/discussion within the same passages and occasionally uses phrasing that implies false equivalence. It could also be made more approachable to a broad readership by adding an overview figure. Given that the manuscript has already undergone one revision where there were comments on both precision of specific statements and comments on general clarity, I would now strongly recommend the recruitment of an experienced (human) scientific proofreader to help improve organization, clarity, and precision.

In terms of speculations, some claims, particularly those relating to mechanisms of biogenesis/assembly, still go beyond what is directly supported by the data. For example, the knockout analyses identify components that are required for assembly, but in my view, they do not establish a mechanism of assembly (where the phrase was removed from the title but remains in the abstract and introduction). In Figure S9, it is not obvious that the arrows highlight the features described in the text.

Finally, the figure quality concerns remain for many/most panels. The text in most (not all) panels is crisp/clear, however, the corresponding graphics in most (not all) panels appear blurry, which suggests that the problem is most likely that the base images for many of the figures were created at low resolution and subsequently up-scaled. (Note, however, that the bar graphs and GSPSC curves do not have crisp text). This is particularly noticeable for the graphics in Fig. 3b (formerly Fig. 1e) and Supplementary Fig. 7a, which now look worse than in the original submission. It will likely be necessary to fully remake the base graphics in these figures, outputting the images at much higher resolution.

Overall, I still consider the work to potentially be of high importance for the field. By focusing the text on only the supported conclusions, improving the organization, increasing language precision, and improving figure quality, the impact of the work could be better conveyed.

Reviewer #1 (Remarks to the Author):

I have reviewed this manuscript before for another journal. This version is improved and the authors have satisfactorily answered my previous comments. Only one point to highlight, I did not follow the authors' answer to point 1 (Reviewer 1). First, line 276 is on page 9 in the version I have and it has nothing to do with the point being raised here (it corresponds to the third line of the discussion in the version I have).

We apologise for this, the page/line numbering must have been confused with the track-change version of the revised manuscript. The corresponding sentence was page 8 line 241-242 (now lines 239-240).

Second, I would strongly encourage the authors to mention the statistics of each observation they describe in the paper, or just make a supplementary table summarizing the number of examples related to each observation they describe and the number of cryo-tomograms. For example, "we looked at XX number of WT *V. cholerae* with TEM and we did not observe empty sheath tips in the filaments", "we looked at XX number of FliDd4 mutant and XX% of them had empty sheath tips", "we collected XX cryo-tomograms of *V. alginolyticus* flhGfliS mutant and saw this number of empty OM extensions", same applies for WT, etc.

We have included this information, in the corresponding figure legends, and in the text of the revised manuscript (page 8, line 239-240; page 10 lines 263-266; and page 14 lines 432-433). In particular, for the tomography experiments, we have collected 36 tilt series ($\Delta fliG$) and 84 tilt series ($\Delta fliG\Delta fliS$), however only few contained intact flagella (4 and 3, respectively). These numbers have been indicated in the corresponding sections, but we did not provide statistics because of these low numbers.

Reviewer #2 (Remarks to the Author):

The authors have substantially rewritten their manuscript and resolved my earlier concerns about overstatements in the interpretation of their results. What remains is solid work, in part confirming earlier findings (e.g. membrane staining of sheath and lack of PG under the sheath was previously established), and in part advancing the field (structure of flagellin, and relevance of FliD D4 for interaction between sheath and filament at tip).

Remaining comments:

1. I agree with reviewer 1 that the authors should give some statistical indication of how common empty sheaths are in the Delta FliD D4 mutant compared to the WT. It is helpful that the manuscript now states that these were not observed at all in the WT, but it would also be helpful if there was information on how common they are in the mutant (how many of how many inspected sheaths seemed empty?).

As indicated in our response above, we have now explicitly added numbers of observations to the figure legend and corresponding sections of the manuscript, to provide a sense of how frequent they are.

2. Is it possible to infer anything about interactions between inner membrane and FliD D4 based on the charge of the latter? Minimally, one would expect that D4 should be less negatively charged than the surface of the flagellin to permit a favorable interaction.

This is a very astute suggestion; we have now included an electrostatic representation for the FliD complex (Supplementary figure 7e), showing that indeed, the region of FliD D4 proposed to form contacts with the membrane is positively charged, unlike the filament surface. This supports our model of FliD D4 forming direct interactions with the sheath.

Minor comments:

1. References appear to be missing for a number of statements if not whole paragraphs, e.g.

1.1. l 52-60

1.2. l. 189-191

1.3. l. 233

2. l. 61-62: *Bdellovibrio* is not a human pathogen as stated.

References have been added accordingly, and the statement on *Bdellovibrio* has been corrected.

Reviewer #3 (Remarks to the Author):

The revised manuscript by Qin et al., now titled “The structure of the *Vibrio alginolyticus* flagellar filament suggests molecular mechanism for the rotation of sheathed flagella,” has been reformatted as a regular article for Nature communications. The authors’ data demonstrate: (1) a structure of the *V. alginolyticus* flagellar filament obtained by cryoEM and cryoET; (2) that FlaD2 is the dominant flagellin in the filament; (3) the physical arrangement of flagellin D2 into protofilaments; (4) how helical interactions between protofilaments generate 11-fold symmetry; (5) that the filament is curved and has a negatively charged exterior; and (6) a tentative assignment for the cap at the flagellar tip. I was the original Reviewer 3 and had initially assumed that some of the claims that did not seem fully supported by the data might reflect the condensed format, and the original review focused on how the selected format limited clarity. I also made comments on additional metrics that would be useful in the structural table, which have been addressed.

In the revised manuscript, which now uses a longer format, the lack of clarity remains (unfortunately) as do the speculations. In terms of clarity, the manuscript is not sufficiently cohesive, and the wording often lacks scientific precision (the other reviewers gave specific examples of lack of precision, but I felt that this was extensive and influenced by the format. It remains extensive and beyond what is reasonable for a reviewer to list one by one). The narrative at times shifts between first- and third-person voice and between active and passive voice. It sometimes conflates

introduction/results/discussion within the same passages and occasionally uses phrasing that implies false equivalence. It could also be made more approachable to a broad readership by adding an overview figure. Given that the manuscript has already undergone one revision where there were comments on both precision of specific statements and comments on general clarity, I would now strongly recommend the recruitment of an experienced (human) scientific proofreader to help improve organization, clarity, and precision.

We apologise for this view by this reviewer, however without any details of which aspect(s) to modify, we cannot address specific issues. An overview figure is provided in figure 6 in the revised manuscript. We reiterate that this manuscript was written without the help of generative AI as implied here.

In terms of speculations, some claims, particularly those relating to mechanisms of biogenesis/assembly, still go beyond what is directly supported by the data. For example, the knockout analyses identify components that are required for assembly, but in my view, they do not establish a mechanism of assembly (where the phrase was removed from the title but remains in the abstract and introduction).

We have removed the term “assembly” from the abstract and introduction. We have kept it in the result section corresponding the data on FliD, as the role of this protein is to promote filament assembly. We have also kept it in the discussion, as we speculate the implications of our data to understand the assembly of sheathed flagella.

In Figure S9, it is not obvious that the arrows highlight the features described in the text.

We have increased the contrast in these images, to make it clearer where the highlighted features are. We have also included a movie showing the reconstituted tomogram, which was deposited to the EMDB. We think that collectively, these clearly illustrate our interpretation of this data.

Finally, the figure quality concerns remain for many/most panels. The text in most (not all) panels is crisp/clear, however, the corresponding graphics in most (not all) panels appear blurry, which suggests that the problem is most likely that the base images for many of the figures were created at low resolution and subsequently up-scaled. (Note, however, that the bar graphs and GSPSC curves do not have crisp text). This is particularly noticeable for the graphics in Fig. 3b (formerly Fig. 1e) and Supplementary Fig. 7a, which now look worse than in the original submission. It will likely be necessary to fully remake the base graphics in these figures, outputting the images at much higher resolution.

We apologise for this, which must be due to the file conversion during manuscript submission; all figures appear crisp on our end. We will ensure high resolution for the published version of this manuscript.

Overall, I still consider the work to potentially be of high importance for the field. By focusing the text on only the supported conclusions, improving the organization, increasing language precision, and improving figure quality, the impact of the work could be better conveyed.